# Recombinant Subunit Vaccine Candidate against the Bovine Viral Diarrhea Virus

**DOI:** 10.3390/ijms25168734

**Published:** 2024-08-10

**Authors:** Verónica Avello, Santiago Salazar, Eddy E. González, Paula Campos, Viana Manríque, Christian Mathieu, Florence Hugues, Ignacio Cabezas, Paula Gädicke, Natalie C. Parra, Jannel Acosta, Oliberto Sánchez, Alaín González, Raquel Montesino

**Affiliations:** 1Biotechnology and Biopharmaceuticals Laboratory, Pathophysiology Department, School of Biological Sciences, Universidad de Concepción, Víctor Lamas 1290, Concepción P.O. Box 160C, Chile; veavello@udec.cl (V.A.); sasalazarmh@gmail.com (S.S.); pcampos2016@udec.cl (P.C.); vianamanriquesuarez@gmail.com (V.M.); janacosta@udec.cl (J.A.); 2Department of Medicine, Division of Gastroenterology, Miller School of Medicine, University of Miami, Miami, FL 33146, USA; eeg76@med.miami.edu; 3Virology Section of the SAG’s Sub-Department Network of Animal Health Laboratories, Lo Aguirre, Santiago de Chile 9020000, Chile; christian.mathieu@sag.gob.cl; 4Pathology and Preventive Medicine Department, School of Veterinary Sciences, Universidad de Concepción, Vicente Méndez 595, Chillán P.O. Box 537, Chile; flohugues@udec.cl (F.H.); oscabeza@udec.cl (I.C.); pgadicke@udec.cl (P.G.); 5Pharmacology Department, School of Biological Sciences, Universidad de Concepción, Victor Lamas 1290, Concepción P.O. Box 160C, Chile; osanchez@udec.cl; 6Faculty of Basic Sciences, University of Medellin, Cra. 87 No 30-65, Medellin 050026, Colombia

**Keywords:** bovine viral diarrhea virus, subunit vaccine, BVDV subgenotypes, E2 glycoprotein

## Abstract

Multivalent live-attenuated or inactivated vaccines are often used to control the bovine viral diarrhea disease (BVD). Still, they retain inherent disadvantages and do not provide the expected protection. This study developed a new vaccine prototype, including the external segment of the E2 viral protein from five different subgenotypes selected after a massive screening. The E2 proteins of every subgenotype (1aE2, 1bE2, 1cE2, 1dE2, and 1eE2) were produced in mammalian cells and purified by IMAC. An equimolar mixture of E2 proteins formulated in an oil-in-water adjuvant made up the vaccine candidate, inducing a high humoral response at 50, 100, and 150 µg doses in sheep. A similar immune response was observed in bovines at 50 µg. The cellular response showed a significant increase in the transcript levels of relevant Th1 cytokines, while those corresponding to the Th2 cytokine IL-4 and the negative control were similar. High levels of neutralizing antibodies against the subgenotype BVDV1a demonstrated the effectiveness of our vaccine candidate, similar to that observed in the sera of animals vaccinated with the commercial vaccine. These results suggest that our vaccine prototype could become an effective recombinant vaccine against the BVD.

## 1. Introduction

Vaccination has been widely used to control the bovine viral diarrhea disease (BVD), which generates significant economic losses worldwide, estimated at more than USD 70 million in dairy and beef cattle farms [1,2,3]. Since the beginning of vaccination against the bovine viral diarrhea virus (BVDV) more than five decades ago [1], the evolution of those vaccines has provided valuable insights into effectively managing the disease. Two main challenges must be addressed to control or avoid the disease through vaccination: the high genetic and antigenic variability of BVDV viral subgenotypes [2,3,4] and viral fetal infection of pregnant cows [5].

Initially, monovalent inactivated vaccines (IVs) and modified-live vaccines (MLVs) did not offer proper protection because of the antigenic divergence between BVDV1 and BVDV2 genotypes. The generation of multivalent IVs and MLVs containing both viral subtypes constituted a step ahead for BVD prevention by increasing cross-protection among different viral subgenotypes [6]. It is demonstrated that MLVs induce neutralizing antibodies against up to 20 antigenically different strains of BVDV1 and BVDV2 [7,8]. Cross-reactions have also been observed during cattle vaccination with IVs, which induce equivalent neutralizing antibodies to MLVs against diverse strains of BVDV2 [9].

On the other hand, the viral infection of fetuses during pregnancy could lead to the birth of persistently infected (PI) animals, which are considered the main source for spreading the BVDV [3]. If fetal infection occurs in the first three months of pregnancy, when the bovine immune system is still immature [10], the offspring would be PI. These animals cannot discriminate virally from self-antigens, resulting in a persistent infection that can be spread to susceptible animals through feces, nasal discharge, tears, and other secretions of PI animals [11]. Both conventional vaccines (MLVs and IVs) have demonstrated fetal protection [12,13]. In a study using an IV, PI animals were eliminated when circulating, and vaccine strains belonged to the same subgenotype [14]. However, when the subgenotype of the field strain was different from that of the IV, the protection against PI offspring was unsuccessful [15]. The consensus is that MLVs induce a more potent humoral and cellular immune response against BVDV than IVs, providing a higher degree of protection from clinical symptoms and long-lasting immune responses [13,16,17].

Unfortunately, conventional vaccines against BVDV have significant drawbacks, including the reversal of virulence and the generation of PI offspring attributed to MLVs. Moreover, IVs have a short duration of the protective immune response, and the stimulation of cellular response could be hampered [18]. When these vaccines are used, it becomes complicated to differentiate infected from vaccinated animals [11,19].

New vaccine approaches may overcome the challenges posed by conventional vaccines. The E2 structural glycoprotein is the main immunogenic antigen of BVDV and induces neutralizing antibodies in the animal host after natural viral infection. Therefore, several studies have used this molecule as a vaccine candidate. The E2 glycoprotein is a type 1 transmembrane protein involved in host receptor binding, such as ADAM17 and CD46. The sequence of this viral envelope protein shares a high degree of similarity across different pestivirus species (about 60–70%) [20]. Recombinant vaccine candidates have been engineered based on the glycoprotein E2, inducing high titers of neutralizing antibodies in vaccinated animals [21,22]. Several studies have evaluated DNA vaccines to counteract BVDV. A combination of DNA sequences from the E2 protein with other viral proteins, activators of the innate immune response, molecular adjuvant, and distinct DNA administration routes demonstrated the induction of effective immune responses in cattle and mice [23,24,25,26].

Additionally, relevant studies with subunit vaccine candidates against BVDV using the E2 protein have suggested a positive contribution to the veterinary health problem caused by this virus. The E2 protein of BVDV has been obtained in mammalian, insect, and plant cells, with encouraging results as a vaccine candidate in mice, goats, and cattle [27,28,29,30,31]. Targeting E2 to dendritic cells or the MHC class II antigen epitope has demonstrated animal protection upon challenge [30,32].

Recently, we constructed a chimeric protein containing two E2 molecules from distinct subtypes and a dimerization domain, rendering cross-neutralization with sera from heterologous subtypes [33]. Occasionally, the neutralizing antibody levels raised against subtypes other than homologous are insufficient and do not correlate with protection against heterologous challenges. More effective immune responses have been observed when circulating and vaccine strains have the same viral subtype.

In this study, we developed a recombinant vaccine candidate against BVDV based on the viral subgenotypes identified through an extensive cattle survey from the most representative farms in Chile and relevant previous studies [34,35]. The E2 sequences of five subgenotypes were cloned and expressed in mammalian cells. After purification, an equal amount of the five E2 proteins was formulated in Montanide ISA 61 VG to further test it as a vaccine candidate in vitro and in vivo.

## 2. Results

### 2.1. BVDV Viral Survey

To develop an effective subunit vaccine against the BVDV, accurate detection of the circulating viral strains is required. The similarity degree between the circulating viral strain(s) and the vaccine’s active components directly influences the immune response’s efficacy against the virus.

Our research encompassed a comprehensive viral survey from January 2019 to December 2022 in relevant BVDV-infected farms from the Maule to Magallanes regions in Chile. Additionally, we screened samples from the SAG serum library in Osorno to expand our dataset. The main objectives were to determine the current BVDV incidence and the circulating subgenotype(s) to design and produce a subunit vaccine candidate.

A total of 1955 serum samples were tested using ELISA to detect anti-BVDV antibodies. Positive results were observed in 1636 samples, while 302 were negative for antibodies and 17 showed ambiguous results. The presence of antibodies could be due to either infection or vaccination, making it difficult to distinguish between infected and vaccinated animals for genotyping and determining the true viral incidence. This difficulty arises because the conventional vaccine CattleMaster^MR^ Gold FP 5 was used for vaccination. Therefore, samples that tested negative for BVDV antibodies were used for viral detection. We identified 132 positive samples (43.7% of antibody-negative samples) with ELISA against BVDV antigen detection, indicating the presence of PI animals. Subsequent sequencing of 31 samples showed that around 80% corresponded to BVDV1b, with the remaining 20% attributed to the BVDV1d subgenotype (Table 1).

Recently, BVDV1b (25%), BVDV1d (8.3%), and BVDV1e (66.7%) were reported during a screening conducted on a lot of animals from the Aysén region in 2017. This was Chile’s first report of the subgenotypes BVDV1d and BVDV1e [34]. Additionally, the subgenotypes BVDV1a, BVDV1b, and BVDV1c were previously reported in Chile between 1993 and 2001 [35]. Considering the above information, we decided to include the five subgenotypes previously detected (BVDV1a, BVDV1b, BVDV1c, BVDV1d, and BVDV1e) into the subunit vaccine candidate.

### 2.2. Production of E2 Proteins from Different BVDV Subgenotypes

The E2 proteins were expressed in CHO-K1 cells after being transfected with DNA encoding the sequences of the five E2 proteins. The fluorescence activity of green fluorescent protein was used as a flow cytometry marker to select high-producing clones from the transfected CHO-K1 cells. The selected clones exhibited high fluorescence intensity within their cells in the culture (Figure 1).

The E2-expressing clones were amplified, and E2 proteins secreted into culture supernatants were purified by IMAC (Appendix A). Western blot assays immunoidentified the five E2 proteins after purification using fluorescent secondary antibodies, where the 6xHis tag and the five specific tags included in the design for expressing each E2 protein were individually visualized (Figure 2).

### 2.3. Dose Response Evaluation in a Sheep Model

The purified E2 proteins were formulated in equal quantities with the oily adjuvant Montanide ISA 61 VG at different doses to assess their immunogenicity in an in vivo experiment carried out in sheep (Figure 3). We decided to use this animal model because of the humoral response correspondence between sheep and bovines using inactivated BVDV vaccines [36]. Antibody levels in animals immunized with three different doses of the vaccine candidate (50 µg, 100 µg, and 150 µg) became notorious at day 14 after the first immunization, with absorbance (OD) values over 0.39. These levels increased on day 21 and after the booster on day 28. The antibody levels remained relatively stable until day 42, when a slight decrease was observed. There were no significant differences among the three doses, except for days 28 and 35, where OD values of the 150 µg dose were significantly higher than the 100 µg dose.

The experiment showed no variations in sheep’s appetite and water consumption. The sheep’s weight stayed stable seven days after the first immunization, and we did not notice any redness, swelling, or pain at the injection site. The difference in average local temperature between the right buttock (injection site) and the left buttock (control) was generally within 5 °C in most measurements, and an average temperature of 0.06 °C was recorded (Appendix A). Although a significant difference in the mean temperatures was detected on day 2, it was not observed at other times during the evaluation period. Therefore, the different temperature values followed a weather pattern of the evaluation period rather than a specific effect of the vaccine candidate inoculation. Usually, the ovine corporal temperature ranges between 38.3 and 40 °C [37,38]. Rectal temperatures remained within the expected ranges for the species in most of the experiment. On day 6, there was only a significant difference between the negative control and the experimental group of 100 µg dose (Appendix A). However, the latter was in the upper limit of the expected temperature, and the highest dose (150 µg) did not have significant differences compared with the other groups. Therefore, we do not think this difference corresponds to an immunization reaction.

Overall, the three doses applied induced a robust humoral immune response, and the antibody levels of the lowest dose (50 µg) did not differ significantly from the other two doses at any time assessed (Figure 3). Although there is a remarkable difference in size between sheep and cows, we decided to continue the immunogenicity experiment in the target species with the lowest dose (50 µg).

### 2.4. Immune Response Induced by the Vaccine Candidate in Bovines

#### 2.4.1. Humoral Immune Response

The humoral immune response in cattle considered three experimental groups: the negative control (NC), the vaccine candidate at a dose of 50 µg (VC), and the commercial vaccine CattleMaster^MR^ Gold FP 5 (CVCM) (Figure 4A). OD values were significantly higher on days 14 and 21 only in the VC group. After the booster, an OD increase was observed in the CVCM and the VC groups, which was sustained until day 35.

From day 42 till the end of the experiment, there was a faint OD decrease for both experimental groups, but the OD values were always above 0.6. After the booster, there were no significant differences between the CVCM and the VC groups for all the times evaluated, but the latter group showed higher OD at all assayed times. The control group did not show specific antibodies against BVDV E2 protein during the experiment (Figure 4B).

Antibody titration was performed using the animal sera of day 56 from the CVCM and the VC experimental groups (Figure 4C). There were no significant differences in the titers of both groups.

The contribution of every E2 antigen of the vaccine candidate to the humoral immune response was also determined (Figure 4D). All E2 proteins of the different BVDV1 subtypes included in the vaccine candidate induced a humoral immune response. The highest immune response was detected against the BVDV1a subtype, which was significantly different from BVDV1b, BVDV1c, and BVDV1d. The lowest contribution to the immune response corresponded to the BVDV1d subtype. The humoral immune response detected when the antigen mix was used reached an average value among antigens with the highest and lowest representation in the humoral immune response.

In this trial, we also recorded rectal temperatures three days after the first inoculation (Appendix A). Temperature values fluctuated in the expected range for the species when comparing all individuals, and no significant differences were registered between groups.

#### 2.4.2. Cellular Immune Response

Transcript levels of IFN-γ, IL-12, and IL-4 were measured using real-time PCR to assess the cellular immune response. On day 21 after the initial immunization, the IFN-γ transcript levels were almost undetectable for the CVCM and the VC experimental groups (Figure 5A). However, on day 42, those levels significantly increased in the VC group. By day 56, both experimental groups significantly increased the IFN-γ transcripts. Regarding IL-12 transcript levels, only the VC group showed a significant increment at all time points evaluated (Figure 5B). The IL-4 transcript levels were not detected (Figure 5C).

The IFN-γ secretion into the cell supernatant of PMBC culture was also measured using ELISA at day 56, where a significant increment was detected in both experimental groups compared with the NC (Figure 5D). There were no significant differences between the CVCM and VC groups, neither in the levels of cytokine transcripts nor in the amount of IFN-γ measured by ELISA.

### 2.5. Determination of Neutralizing Antibodies

Neutralizing antibodies were measured by an in vitro neutralization assay using the viral subtype BVDV1a (Figure 6). This specific subgenotype was selected because it shows the highest antibody response, which was significantly different from most of the other subgenotypes when evaluating the individual contribution of every subgenotype (Figure 4D). The vaccine candidate elicited almost the same levels of average neutralizing antibodies compared with the values of the commercial vaccine at both time points analyzed, which were significantly higher than the NC group.

## 3. Discussion

Basic research on new vaccine candidates against BVDV has expanded the perspectives for the control and/or eradication of this virus. Several studies have succeeded in inducing proper immune responses and protecting animals upon challenge with the pathogen using immunogenic formulations other than the traditional ones. DNA vaccines encoding the sequence of the E2 viral protein alone [40,41] or in combination with sequences of different molecules, such as the pattern receptor retinoic acid-inducible gene I (RIG-I), interleukin-2 (IL-2), and granulocyte-macrophage colony-stimulating factor (GM-CSF), have demonstrated the induction of virus-specific neutralizing antibodies against homologous and heterologous strains, Th1 like lymphocyte proliferation, and partial protection in mice and cattle [25,42]. The electroporation of DNA vaccine candidates using the TriGrid™ Delivery System for intramuscular delivery (TDS-IM) induces rapid and effective immune responses, with a significant increase in neutralizing antibody titers and the number of activated T cells. Also, viral shedding and clinical signs have been reduced [24,43]. Good results were also obtained when immunization schemes combined DNA prime-protein boost with the E2 DNA sequence/protein. E2-specific antibody titers were increased, lymphocytes secreting IFN-γ were detected, and protection was achieved in mice and cattle. Little leukopenia, significant viremia reduction, minor pathological changes, and steady animal weight and temperature were recorded [44,45].

Additionally, many studies have been conducted to evaluate different recombinant subunit vaccine candidates against BVDV. The E2 protein is usually considered the main viral immunogen. It has been produced as a vaccine candidate in several expression systems, such as bacteria [32], plants [29,46], insects [28,30,47,48,49,50], and mammalian cells [31,51,52,53,54], inducing an adequate immune response and protection upon challenge in Guinea pigs, mice, sheep, and cattle. Chimeric constructions based on the E2 protein fused to a DC-targeting peptide, the Fc fragment of the bovine IgG1, the Helicobacter pylori ferritin, and a single-chain antibody targeting the major histocompatibility type II molecule (MHC-II) have shown the generation of specific neutralizing antibodies and conferred distinct protection degrees to immunized animals depending on the viral strain used for the challenge [30,32,54].

The multiantigenic concept using subunit vaccine candidates against BVDV has also been managed, and encouraging results have been obtained [50,51]. The great diversity of this virus, having two main genotypes (BVDV1 and BVDV2) and at least 25 subgenotypes [3] with demonstrated genetic and antigenic differences [55], makes the design of effective vaccines challenging. Cross-neutralization among BVDV strains varies considerably [56,57], making the homologous challenge more protective post-vaccination than the heterologous challenge [58,59,60]. These factors highlight the complexity involved in developing vaccines that provide broad protection against the diverse BVDV strains. The analysis of circulating viral strains could help to design better vaccines for BVDV control. Hence, we conducted a field survey in this study and considered current and previous circulating strains to design and produce a multivalent vaccine candidate. The aim was to extend the protection range against several South American BVDV endemic subgenotypes.

The ectodomain of E2 proteins corresponding to subgenotypes BVDV1a, BVDV1b, BVDV1c, BVDV1d, and BVDV1e were cloned and produced in mammalian cells to create a novel pentavalent subunit vaccine candidate. This vaccine induced a humoral immune response characterized by high antibody titers after boosting, and each of the five E2 antigens included in our vaccine candidate showed immunogenicity. The immune response elicited by our vaccine candidate was compared to a commercial vaccine (containing subgenotype 1a and genotype 2). Both vaccines showed neutralizing antibody values over 1/256 when using the BVDV1a subtype for the neutralization assay, and no significant differences were observed between these experimental groups. Controversial results can be found in the scientific literature when trying to correlate the signs of animal protection and values of neutralizing antibodies. A study established that antibody titers did not predict the protection level for immunized steers [61]. However, some research has shown noticeable protection levels with neutralizing titers above 1/512 [62], while titers of 1/256 or above can counteract clinical symptoms [63,64]. It seems that the protection concept is also associated with cellular immunity because animals with low antibody levels were protected after a viral challenge [65]. In this study, a significant increase in the transcription and translation of IFN-γ was observed, suggesting a Th1 cellular response. Evidence provided here indicates that our subunit vaccine candidate could potentially provide accurate protection upon a homologous challenge against the subgenotype BVDV1a, according to the neutralization assay.

## 4. Materials and Methods

### 4.1. Cell Lines and Animals

The CHO-K1 cell line (ATCC^®^ CCL-61) was used as the host for producing recombinant E2 proteins. In vivo experiments complied with national guidelines and the authorization of the Ethical Committee of the University of Concepcion (CEBB-1054-2021). A total of 24 female Suffolk Down sheep, nine months old and weighing around 50 kg, were used to evaluate the vaccine candidate. Before the experiment, animals were tested with the ELISA kit IDEXX BVDV Total Ab (IDEXX, Westbrook, ME, USA) to ensure they were seronegative to BVDV. The feeding regime was based on forage and concentrate, equivalent to 3% of the weight. Fresh water was supplied ad libitum. Twenty-four Red Friesian cattle, eight months old and with an average weight of 240 kg, were purchased from private production farms. They were serologically negative for BVDV. Animals were fed with mixed clover hay, ryegrass, and a 21% protein-concentrated food, reaching an equivalent of 3% body weight. Fresh water was supplied ad libitum.

### 4.2. BVDV Subgenotype Screening

Viral screening was performed using serum samples from infected farms in different regions of Chile. We also used the Agricultural and Livestock Service (SAG) serum library samples. Serum samples were evaluated using the ELISA IDEXX BVDV Ab/serum and ELISA IDEXX BVDV Ag/serum (IDEXX, Westbrook, ME, USA), as well as by DNA amplification using the following primers based on those proposed by Vilcek et al. 1994 [66]: Forward: 5′ CATGCCCWTAGTAGGACTAGC 3′, reverse: 5′ WCAAYTCCATGTGCCATGTAC 3′, where W means A or T and Y means C or T. Reverse transcription was performed with the RevertAid First Strand cDNA Synthesis kit (Thermo Scientific, Waltham, MA, USA) as follows: 5 min at 25 °C, 1 h at 42 °C, and 5 min at 70 °C. The cDNA was amplified by PCR using the Platinum SuperFI II PCR Master Mix kit (Invitrogen, Waltham, MA, USA) and the automatic thermocycler Tprofessional Basic 96 (Biometra, Germany) at the following temperature cycle: 1 min at 94 °C, 45 cycles of 5 s at 98 °C, 10 s at 55 °C, and 5 s at 72 °C. Three minutes at 72 °C were programmed at the end. PCR products were sequenced by Macrogene, Seoul, Republic of Korea.

### 4.3. Generation of CHO-K1 Clones Expressing Different E2 Proteins

The CHO-K1 cell line was genetically modified to produce E2 proteins corresponding to the five BVDV subgenotypes previously selected. Gene sequences of bicistronic transcriptional units coding the E2 proteins attached to distinct tags for immunoidentification and the 6-His tag (1aE2-cMyc-6xHis, 1bE2-HA-6xHis, 1cE2-VSV-6xHis, 1dE2-V5-6xHis, and 1eE2-E-6xHis), followed by an internal ribosome entry site (IRES) and the fluorescent green protein (GFP), were chemically synthesized. They were inserted in the mammalian expression vector pCI-neo by the company GenScript, Piscataway NJ, USA. The resulting plasmids were named pC1-1aE2-LeGL, pCI-1bE2-LeGL, pCI-1cE2-LeGL, pCI-1dE2-LeGL, and pCI-1eE2-LeGL. Transcriptional units were controlled using the cytomegalovirus promoter/enhancer (CMV) and the SV40’s transcription termination sequence. Proteins were directed into the secretory pathway using the human albumin signal peptide, and the individual clone selection was performed using the Neomycin/Kanamycin marker (Figure 7). Plasmids were transformed and amplified in One Shot™ TOP10 Chemically Competent *E. coli* cells (Thermo Fisher Scientific, Waltham, MA, USA). The plasmid purification was performed using the alkaline lysis method described by Birnboim and Doly in 1979 [67]. Every plasmid was linearized with the restriction enzyme *BamH1* (New England BioLabs, Ipswich, MA, USA) and cleaned using the GenElute PCR Clean-Up kit (Sigma-Aldrich, St. Louis, MO, USA) for further transfection into CHO-K1 cells using Lipofectamin 2000 (Invitrogen, Waltham, MA, USA) according to the manufacturer’s suggestions. The clone selection was performed by measuring the fluorescence intensity emitted by GFP in a flow cytometer (BD FACSAria cytometer. III Cell Sorter, Franklin Lakes, NJ, USA). At least 10,000 events were counted, and clones with the highest fluorescence intensities were chosen by cell sorting for further individual culture in 96 well plates (Thermo Fisher Scientific, Waltham, MA, USA).

### 4.4. Purification of E2 Proteins

Selected clones were amplified in RPMI medium (HyClone, Marlborough, MA, USA) with 10% fetal bovine serum (FBS) and 800 µg/mL G418 at 37 °C, 5% CO_2_, and 95% relative humidity until confluence. Supernatant recovery was performed after 48 h of incubation with RPMI medium without FBS, which was stored at –20 °C until use. The E2 proteins contained in culture supernatants were purified by Immobilized Metal Affinity Chromatography (IMAC) using a matrix of Nickel-sepharose high performance (Sigma-Aldrich, St. Louis, MO, USA) and an AKTA Prime Plus chromatography station (Cytiva, Marlborough, MA, USA). Samples were entered into the column at 5 mM imidazole, washed with 50 mM imidazole, and eluted with 400 mM. After chromatography, eluates were dialyzed in a phosphate buffer solution (PBS) for 16 h at 4 °C under slow agitation and concentrated by adding PEG 35,000 (Santa Cruz Biotechnology, Dallas, TX, USA). Protein quantification was performed by combining the BCA total protein determination kit (Thermo Fisher Scientific, USA) with densitometric analysis in the Image Studio software version 3.1 and the ODYSSEY CLx imaging system (Li-Cor, Lincoln, NE, USA) after SDS-PAGE in a 12.5% gel. Multiple BSA (Thermo Fisher Scientific, Waltham, MA, USA) concentrations were used as standard.

### 4.5. SDS-PAGE and Western-Blot

SDS-PAGE analysis was performed as described by Laemmli in 1970 [68] using 12.5% gels. For Western blots, proteins were transferred to nitrocellulose membranes of 0.45 µm (PerkinElmer, Shelton, CT, USA) using a semi-dry electroblotting TransBlot-Turbo (Bio-Rad, Hercules, CA, USA) at 0.3 A and 25 V for 30 min. After blocking with 5% skimmed milk in PBS, primary antibodies for every tag were added as corresponded, together with the anti-6xHis antibody. Secondary antibodies were added according to the specifications of primary antibodies. Infrared signals were detected using the ODYSSEY CLx imaging system and the Image Studio software version 3.1 (Li-Cor Biosciences, Lincoln, NE, USA). All antibodies used in Western blot assays are listed below (Table 2).

### 4.6. Vaccine Formulation

The vaccine candidate was formulated with equivalent quantities of purified E2 proteins (1/5 each E2 protein in the total aqueous volume) and Montanide ISA 61 VG (SEPPIC, Paris, France) as an adjuvant. An aqueous/oil phase ratio of 60/40 was used. Placebo formulations were made with PBS.

### 4.7. Evaluation of the Vaccine Candidate in Sheep

Sheep were gathered into four experimental groups of six animals each: Group 1: placebo, Group 2: vaccine candidate at 50 µg/mL, Group 3: vaccine candidate at 100 µg/mL, and Group 4: vaccine candidate at 150 µg/mL. Formulations (vaccine candidate and placebo) were administered intramuscularly at 1 mL volume. The immunization scheme included a first immunization on day zero and a second on day 21. Blood samples were collected on days 0, 14, 21, 28, 35, and 42. The humoral immune response was evaluated by indirect ELISA. The inoculation site was also monitored by measuring redness, pain, and local temperature. This last parameter was adjusted by measuring the temperature of the contralateral gluteus. Systemic parameters, such as transrectal body temperature, pathologic alterations, respiratory disturbance, diarrhea, change in mucosal color, and changes in behavior and appetite, were also registered.

### 4.8. Evaluation of the Vaccine Candidate in Bovines

Cattle were randomly distributed into three experimental groups of eight animals each. Two doses of 2 mL were applied on days 0 and 21 by a deep intramuscular injection. The experimental groups were the following: Group 1: Placebo, Group 2: Commercial vaccine CattleMaster^MR^ Gold FP 5 (Zoetis, Parsippany, NJ, USA), and Group 3: 50 µg of the vaccine candidate. After immunization, total blood samples were collected on days 14, 21, 28, 35, 42, 49, and 56. Blood samples were collected in tubes with a coagulation activator (BD, Franklin Lakes, NJ, USA) and centrifuged at 1600× *g* for 8 min. Sera were stored at −20 °C for further evaluation. Additionally, blood samples were collected in tubes containing EDTA as an anticoagulant (BD, Franklin Lakes, NJ, USA) on days 21, 42, and 56 post-immunization to evaluate the cellular immune response.

### 4.9. Evaluation of Humoral and Cellular Immune Responses

#### 4.9.1. Humoral Immune Response

The time course of the humoral immune response was determined by indirect ELISA. Maxisorp™ Nunc plates (Thermo Fischer Scientific, Waltham, MA, USA) were coated with purified E2 proteins (4 µg/mL per protein) in a coating buffer (50 µL/well) for 13 h at 4 °C. To evaluate the contribution of every antigen, wells were coated with 0.5 µg/well of individual antigens. The blocking was performed with 3% skimmed milk in PBS for 1 h at 37 °C. Diluted serum samples of immunized animals (1:1000 for sheep sera and 1:200 for cattle sera) were added (50 µL/well) and incubated for 1 h at 37 °C. The antibody titration was performed using three-fold diluted sheep sera (1:1000 to 1:243,000) and two-fold diluted cattle sera (1:400 to 1:51,200). For individual antigenic contribution, sera from day 56 at 1:200 was used. After two washes with PBS-Tween 0.05% (PBST), a rabbit anti-sheep IgG H&L (HRP) polyclonal antibody (ab6747, Abcam, Waltham, MA, USA) diluted 1:20,000 or a sheep anti-Cow IgG heavy chain (HRP) polyclonal antibody (ab112618 Abcam, Waltham, MA, USA) diluted 1:20,000 was added as a secondary antibody (50 µL/well), which were incubated for 1 h at 37 °C. Plates were washed three times with PBST, and the reaction was developed with the OPD substrate (300 µL/well for sheep sera and 50 µL/well for cattle sera). The reaction was stopped with H_2_SO_4_ 2M (25 µL/well), and the absorbance was measured at 492 nm using a Sinergy^®^ HTK plate reader (BioTek, Agilent Technologies, Santa Clara, CA, USA). For the titration of cattle sera, the cut-off value was determined as the mean absorbance of the negative control plus two-fold the standard deviation of the negative control: Cut-off = (Xneg + 2SDneg). Titer values were plotted as 1/dilution.

#### 4.9.2. Isolation of Peripheral Blood Mononuclear Cells

Separation of peripheral blood mononuclear cells (PBMCs) from the whole blood was performed using Lymphocyte Separation Medium (Corning, Corning, NY, USA) according to the manufacturer’s instructions. PBMCs were washed with sterile PBS at 500× *g* for 10 min. The pellet was resuspended in 5 mL Erythrocyte Lysis Buffer (0.15 M NH_4_CL (Sigma-Aldrich, St. Louis, MO, USA), 10 mM NaHCO_3_ (Sigma-Aldrich, St. Louis, MO, USA), 0.1 mM EDTA (Sigma-Aldrich, St. Louis, MO, USA), and incubated for 5 min at room temperature (RT). After centrifuging at 300× *g* for 10 min, the supernatant was discarded, and the pellet was washed with 5 mL sterile PBS and 5 mL RPMI medium. The pellet was resuspended in 2 mL RPMI medium supplemented with 10% FBS (Gibco, Waltham, MA, USA) and 1% Streptomycin/penicillin (Genesee Scientific, El Cajon, CA, USA). PBMCs were incubated at 37 °C, 5% CO_2_, and 95% relative humidity.

#### 4.9.3. Stimulation of Peripheral Blood Mononuclear Cells

PBMCs were seeded in a 24-well plate (1.5 × 10^6^ cells/well) and treated with E2 antigens at 20 µg/mL in RPMI. The negative control corresponded to cells cultured in RPMI only, and the positive control corresponded to cells exposed to Concanavalin A (Sigma-Aldrich, St. Louis, MO, USA) at 10 µg/mL. PBMCs were cultured for 24 h at 37 °C, 5% CO_2_, and 95% relative humidity, and treatments were performed in triplicate. Supernatants were stored for ELISA analysis, while pellets were resuspended in TRIzol Reagent (Invitrogen, Waltham, MA, USA) and frozen at −80 °C until use.

#### 4.9.4. Evaluation of Cellular Response Markers Using RT-qPCR

Total RNA, previously extracted with TRIzol Reagent, was quantified, diluted to 100 ng/µL, and treated with DNase I (Thermo Fisher Scientific, Waltham, MA, USA). A total of 500 ng of total RNA from each sample was reverse transcribed using the RevertAid First Strand cDNA Synthesis kit (Thermo Fisher Scientific, Waltham, MA, USA) according to the manufacturer’s instructions. The cDNA was stored at −80 °C for further use. Real-time PCR (qPCR) was made using the KAPA SYBR FAST Kit (Kapa Biosystems, Wilmington, MA, USA) and the AriaMx real-time PCR system (Agilent Technologies, Santa Clara, CA, USA). Cycling conditions were 3 min at 90 °C, followed by 40 repetitions of 90 °C for 5 s and 60 °C for 20 s. The gene GAPDH was amplified in every sample and used as housekeeping. The following primers were used for detecting gene transcripts: IFN-γ: forward: AGGTCATTCAAAGGAGCATGGA, reverse: TGCAGATCATCCACCGGAA; IL-12 beta: forward TTCATCAGGGACATCATCAAACCA, reverse CTGAACACAAAACGTCAGGGAG; IL-4: forward GGCGGACTTGACAGGAATCT, reverse TTCAGCGTACTTGTGCTCGT; GAPDH: forward ACACCCTCAAGATTGTCAGCAA, reverse TCATAAGTCCCTCCACGATGC. The CT values from the unstimulated PBMCs were used as calibrators. The relative quantification of mRNAs was calculated using the 2^−∆∆Ct^ method [39].

#### 4.9.5. ELISA for Detecting IFN-γ from PBMCs

IFN-γ concentration in supernatants of PBMCs was determined using the commercial ELISA Flex: Bovine IFN-γ (HRP) (Mabtech, Cincinnati, OH, USA) following the instruction guide. Briefly, plates were coated with 150 ng/well of mAb-bIFN-γ overnight at 4 °C. The plates were blocked with BSA 0.1% dissolved in PBST for 1 h at RT. Then, 100 µL of the kit standards or PBMCs supernatants from day 56, diluted from 1 to 1/200, were added and incubated for 2 h at RT. IFN-γ was detected by incubating with mAb bIFN-γ-biotin (0.25 µg/mL or 0.5 µg/mL, respectively) for one hour at RT and HRP-streptavidin (1/1000) for another hour at RT. Reactions were developed with 100 µL of TMB substrate (Mabtech, Cincinnati, OH, USA) and stopped with 50 µL of H_2_SO_4_ 0.2 M. Results were read at 450 nm in the plate reader (BioTek, Agilent Technologies, Santa Clara, CA, USA).

#### 4.9.6. Neutralization Assay

The virus neutralization assay (VN) was based on the protocol described by the Manual of Diagnostic Tests and Vaccines for Terrestrial Animals (OIE, 2016) using a 96-well flat-bottom microtiter plate. Samples were tested using the cytopathic strain Singer (BVDV-1a) at 100 TCID_50_ (50% Tissue culture infective dose). Sera were diluted 1/5 in PBS and inactivated at 56 °C for 30 min. Two-fold serial dilutions were mixed with the cytopathic strain from 1:10 to 1:1280. After 1 h of incubation at 37 °C, 5% CO_2_, and 95% relative humidity, 50 µL of a primary bovine embryo kidney cell suspension (400,000 cells/mL) were added to each well. Cells with and without the virus were used as cytopathic and growth controls, respectively. Plates were incubated under the same conditions for 4–5 days and microscopically examined for cytopathic effect. Neutralization was considered for samples with no cytopathic effects at dilutions higher or equal to 1:40.

### 4.10. Statistical Analysis

Statistical analyses were performed with the GraphPad Prism version 10.0.0 for Windows (GraphPad Software, Boston, MA, USA). Data were evaluated using parametric and non-parametric methods depending on the results of the Bartlett test for homogeneity of variances and the Shapiro–Wilk test for normality, both with a confidence level of 95%. Data transformation was performed to improve the assumptions of the statistical tests. When parametric assumptions were acceptable, the statistical significance at different times was determined using the one-way ANOVA followed by the Bonferroni test for multiple comparisons. For non-parametric data distributions, the statistical significance between groups at each time was performed using the Kruskal–Wallis test, followed by the Mann–Whitney test adjusted for multiple comparisons with the Benjamini–Hochberg method. The statistical significance among different times was determined using the Wilcoxon matched-pairs signed rank test. For repeated measures of non-parametric data, the Friedman test and Dunn’s multiple comparison test were performed. The statistical analysis specifications are included in figure legends. Significance was considered for *p* < 0.05.

## 5. Conclusions

Undoubtedly, the BVD poses challenging issues for veterinary medicine. The high diversity of the etiological agent, the lack of protection of immunized animals after confronting heterologous viral strains, and the generation of PI animals are pending subjects to be resolved. In this study, we designed and produced a new subunit vaccine candidate with five E2 proteins from different BVDV subgenotypes. The in vivo results showed the generation of neutralizing antibodies against the subgenotype BVDV1a, suggesting potential protection in vaccinated animals after homologous challenge. An additional characterization must be done to corroborate the induction of neutralized antibodies against the other four subgenotypes in our recombinant vaccine candidate.

## Figures and Tables

**Figure 1 ijms-25-08734-f001:**
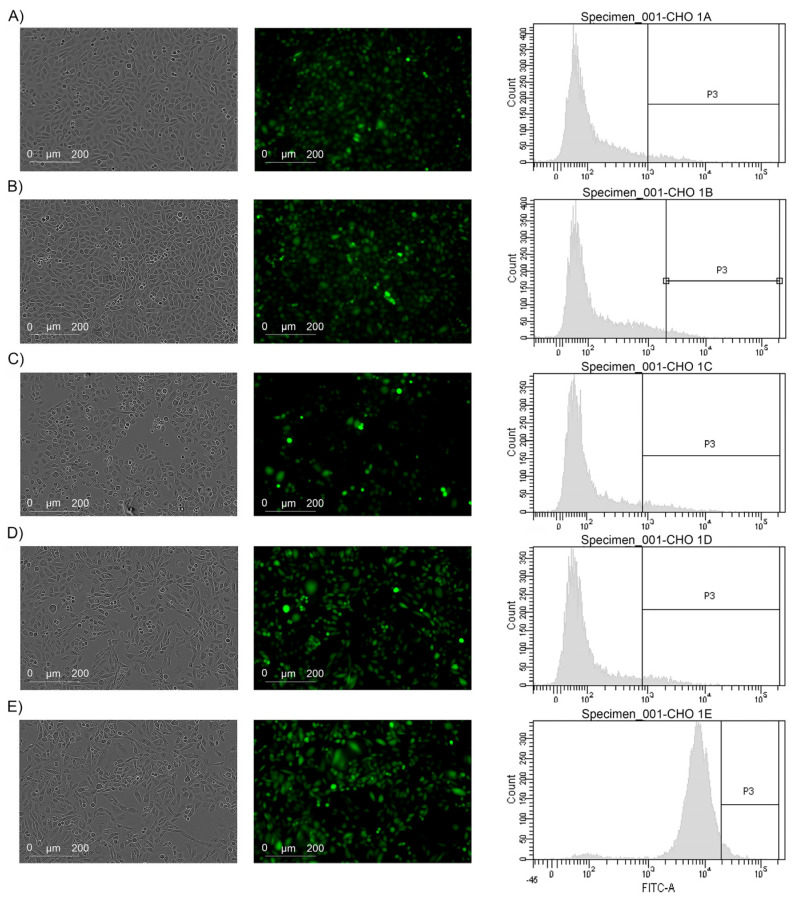
Individual clones expressing different E2 proteins. Bright fields, fluorescences, and flow cytometry histograms from clones expressing (**A**) 1aE2, (**B**) 1bE2, (**C**) 1cE2, (**D**) 1dE2, and (**E**) 1eE2. The P3 region of histograms gathers cells with the highest fluorescence intensity.

**Figure 2 ijms-25-08734-f002:**
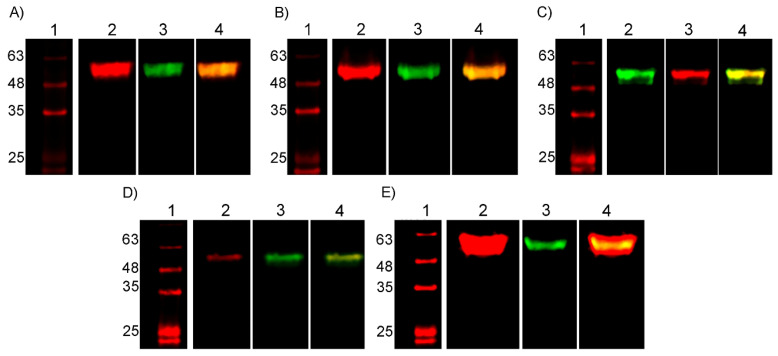
Immunoidentification of the five E2 proteins. Western blot assay of purified E2 proteins using specific primary antibodies against (**A**) c-Myc tag, (**B**) HA tag, (**C**) VSV tag, (**D**) V5 tag, and (**E**) E tag. Anti-6xHis antibody was also used. 1—Molecular Weight Marker AccuRuler RGB Plus prestained protein ladder (Maestrogen, Taiwan), 2—Anti-specific tag, 3—anti-6xHis tag, 4—Merge of anti-specific tag and anti-6xHis tag, meaning they identified the same E2 protein.

**Figure 3 ijms-25-08734-f003:**
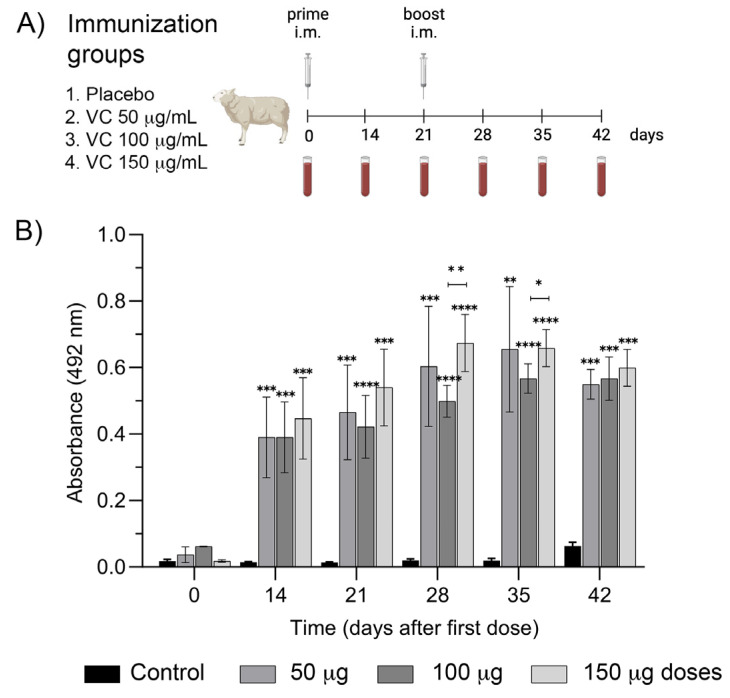
Vaccine candidate’s immunogenicity in sheep. (**A**) Immunization protocol. (**B**) Time-course of humoral immune response in sheep, i.m. immunized on days 0 and 21 with 50 µg, 100 µg, and 150 µg of the vaccine candidate. Data represented means ± SD analyzed by the Friedman test, followed by Dunn’s multiple comparison test, n = 8. * *p* ≤ 0.05, ** *p* ≤ 0.01, *** *p* ≤ 0.001 and **** *p* ≤ 0.0001.

**Figure 4 ijms-25-08734-f004:**
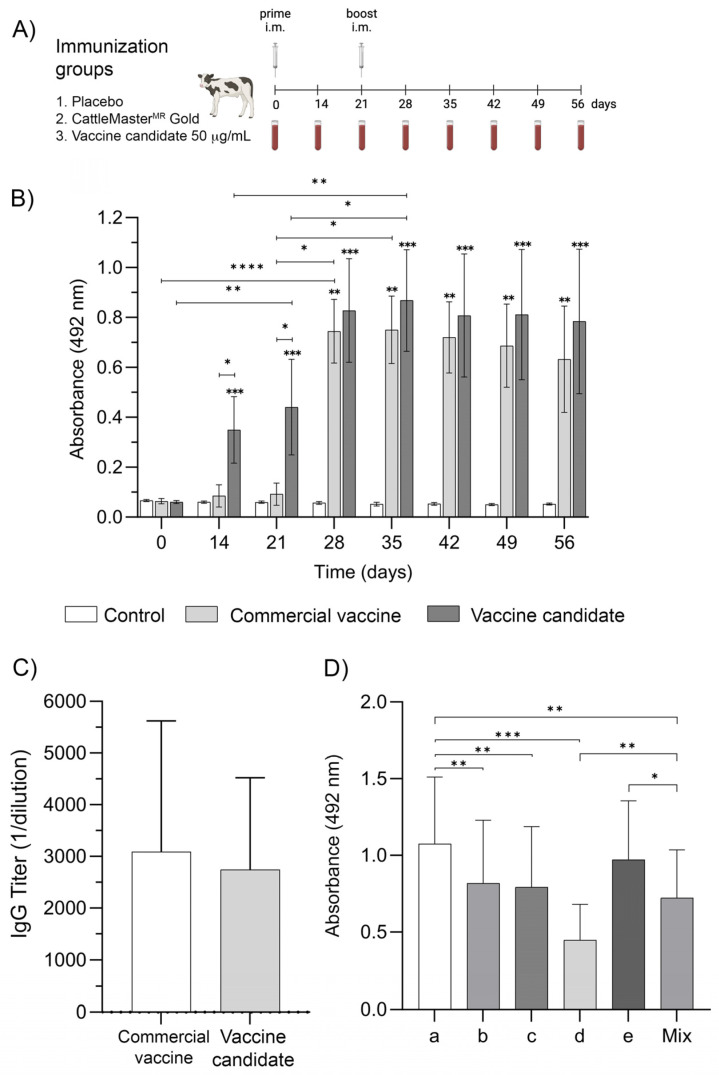
Vaccine candidate’s immunogenicity in bovines. (**A**) Immunization protocol. (**B**) Time course of humoral immune response in cattle immunized with 50 µg of the vaccine candidate. The commercial vaccine CattleMaster^MR^ Gold FP 5 was used as the positive control. Data represent means ± SD analyzed by the Friedman test and Dunn’s multiple comparisons test. The significance between groups at each time was analyzed by the Kruskal–Wallis test and Dunn’s multiple comparisons test. (**C**) Antibody titers in the sera of immunized animals at day 56. Data were analyzed by the Mann–Whitney test. (**D**) The individual contribution of every E2 antigen to the humoral immune response. Data were analyzed by the Friedman test and Dunn’s multiple comparisons test. * *p* ≤ 0.05, ** *p* ≤ 0.01, *** *p* ≤ 0.001 and **** *p* ≤ 0.0001.

**Figure 5 ijms-25-08734-f005:**
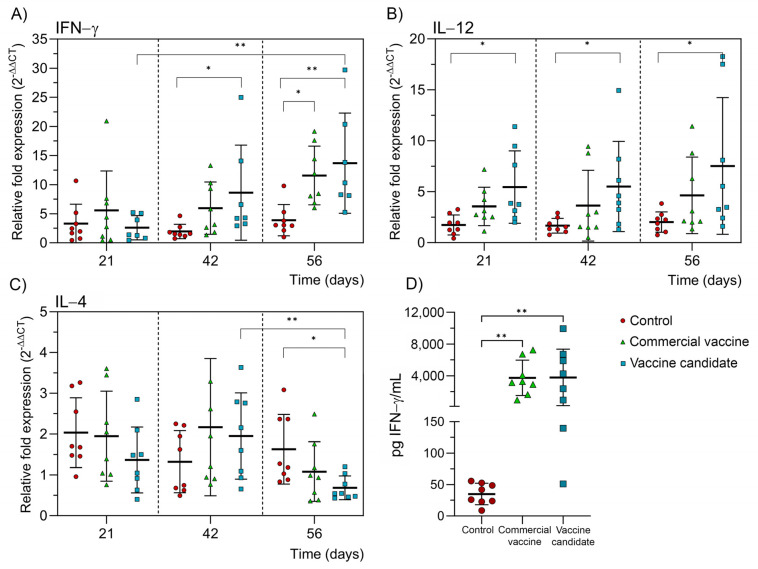
Cellular immune response by measuring cytokine transcripts or protein levels. Relative transcript expression from (**A**) IFN-γ, (**B**) IL-12, and (**C**) IL-4 was measured using the 2^−ΔΔCt^ method [39]. The GAPDH gene was used as housekeeping. (**D**) IFN-γ in vitro determination using unstimulated and stimulated PBMC from day 56. The mean ± SD for each group (n = 8 animals) is represented. Data were analyzed by the Friedman test, followed by Dunn’s multiple comparisons test. Statistical significance between groups at each time was determined by the Kruskal–Wallis test, followed by Dunn’s multiple comparisons test. * *p* ≤ 0.05 and ** *p* ≤ 0.01.

**Figure 6 ijms-25-08734-f006:**
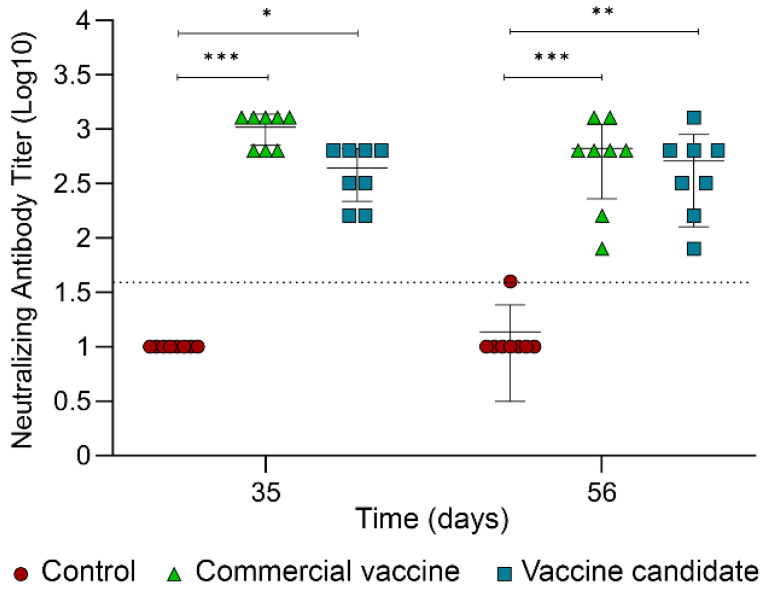
In vitro neutralization assay. Primary bovine embryo kidney cells were treated with a serum-antigen mixture using sera from days 35 and 56 after the first immunization and the reference antigen BVDV1a. Values represent the mean ± SD of 8 animals. Statistics were determined using the Wilcoxon matched-pairs signed rank test for different times. The Kruskal–Wallis test, followed by Dunn’s multiple comparison tests was used to measure the significance between groups at each time. * *p* ≤ 0.05, ** *p* ≤ 0.01, and *** *p* ≤ 0.001.

**Figure 7 ijms-25-08734-f007:**
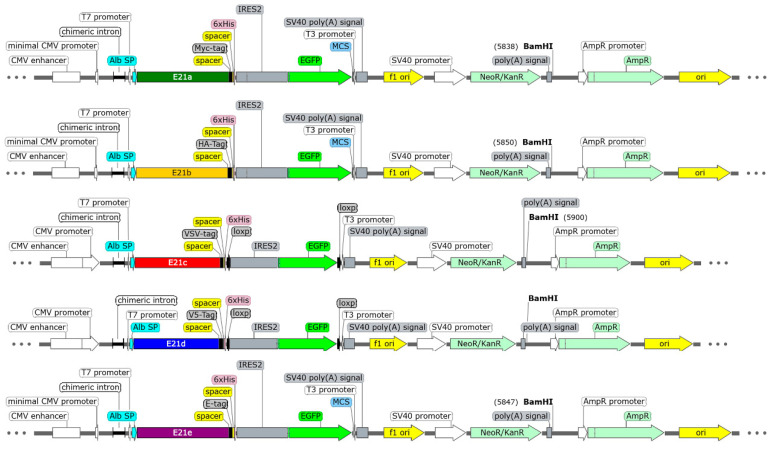
Scheme of the expression vectors coding the five E2 proteins from different BVDV subgenotypes. Bicistronic transcriptional units designed to produce E2 proteins also contain the GFP molecule as a fluorescent marker linked by an IRES sequence. Transcriptional units were controlled using the CMV promoter/enhancer and the SV40 cleavage and polyadenylation sequence. Every E2 protein has a distinct tag (c-Myc, HA, VSV, V5, and E) for immunoidentification. A second tag (6xHis tag) was also included for purification purposes.

**Table 1 ijms-25-08734-t001:** Data were collected from the survey conducted on relevant farms from the Maule to Magallanes in Chile.

Region	Number of Samples	Ab. Positive	Ab. Negative	Ab. Doubtful	Ag. Positive	Seq.	Viral Subtype
Maule	21	0	21	0	21	7	BVDV1b
Ñuble	238	122	110	6	7	7	BVDV1b (6), BVDV1d (1)
Bio-Bio	2	0	2	0	2	0	-
Araucanía	15	0	15	0	15	3	BVDV1b
Los Ríos	58	0	58	0	58	7	BVDV1b (6), BVDV1d (1)
Los Lagos	923	877	38	8	26	7	BVDV1b (3), BVDV1d (4)
Aysén	695	637	55	3	0	0	-
Magallanes	3	0	3	0	3	0	-
Total	1955	1636	302	17	132	31	

**Table 2 ijms-25-08734-t002:** Antibodies used for the detection of E2 proteins in Western blot assays.

Genotypes	Anti-Tag Antibody	Secondary Antibody	Anti-6xHis Antibody	Secondary Antibody
1aE2-cMyc-6xHis	Mouse anti-c-Myc tag, monoclonal antibody (cod. 13-2500, Invitrogen, Waltham, MA, USA). 1/1000	Goat anti-Mouse IgG (H+L) antibody, Alexa Fluor™ 790 (Thermo Fischer Scientific, Waltham, MA, USA) 1/30,000	Rabbit anti-6xHis tag polyclonal antibody (PA1-983B, Invitrogen, Waltham, MA, USA) 1/5000	Goat anti-Rabbit IgG (H+L) antibody Alexa Fluor^®^ 680 (Thermo Fischer Scientific, Waltham, MA, USA) 1/30,000
1bE2-HA-6xHis	Mouse anti-HA tag monoclonal antibody (cod. 26183, Invitrogen, Waltham, MA, USA) 1/5000	Goat anti-Mouse IgG (H+L) antibody, Alexa Fluor™ 790 (Thermo Fischer Scientific, Waltham, MA, USA) 1/30,000	Rabbit anti-6xHis tag polyclonal antibody (PA1-983B, Invitrogen, Waltham, MA, USA) 1/5000	Goat anti-Rabbit IgG (H+L) antibody Alexa Fluor^®^ 680 (Thermo Fischer Scientific, Waltham, MA, USA) 1/30,000
1cE2-VSV-6xHis	Rabbit anti-VSV-G tag polyclonal antibody (cod. A190-131A. Bethyl laboratories. Inc, Montgomery, TX, USA) 1/5000	Goat anti-Rabbit IgG (H+L) antibody Alexa Fluor^®^ 680 (Thermo Fischer Scientific, Waltham, MA, USA) 1/30,000	Mouse anti-6xHis tag monoclonal antibody (cod. 631212, Clontech Laboratories, Waltham, MA, USA 1/5000	Goat anti-Mouse IgG (H+L) antibody, Alexa Fluor™ 790 (Thermo Fischer Scientific, Waltham, MA, USA) 1/30,000
1dE2-V5-6xHis	Mouse anti-V5 tag monoclonal antibody (cod. R960-25, Invitrogen, Waltham, MA, USA) 1/5000	Goat anti-Mouse IgG (H+L) antibody, Alexa Fluor™ 790 (Thermo Fischer Scientific, Waltham, MA, USA) 1/30,000	Rabbit anti-6xHis tag polyclonal antibody (PA1-983B, Invitrogen, Waltham, MA, USA) 1/5000	Goat anti-Rabbit IgG (H+L) antibody Alexa Fluor^®^ 680 (Thermo Fischer Scientific, Waltham, MA, USA) 1/30,000
1eE2-E-6xHis	Goat anti-E tag Epitope Tag polyclonal antibody (NB600-518B, Novus biologicals, Centennial, CO, USA) 1/5000	Rabbit anti-Goat IgG (H+L) antibody, Alexa Fluor™ 790 (Thermo Fischer Scientific, Waltham, MA, USA) 1/30,000	Rabbit anti-6xHis tag polyclonal antibody (PA1-983B, Invitrogen, Waltham, MA, USA) 1/5000	Goat anti-Rabbit IgG (H+L) antibody Alexa Fluor^®^ 680 (Thermo Fischer Scientific, Waltham, MA, USA) 1/30,000

## Data Availability

The authors declare that all data generated in this study are shown in the manuscript and Appendix A.

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
