# Peer review of "Recombinant Subunit Vaccine Candidate against the Bovine Viral Diarrhea Virus"

_ijms, 2024, doi:10.3390/ijms25168734_

Round 1

Reviewer 1 Report

Comments and Suggestions for Authors

The manuscript ‘Recombinant Subunit Vaccine Candidate against the Bovine Viral Diarrhea Virus’ Veronica Avello et.al describes the production and screening of a vaccine generated from the Bovine Viral Diarrhea virus (BVDV) multi-subgenotype E2 protein. The vaccine was screened in multiple representative cattle farms in Chile, and it was also compared to an existing conventional vaccine that is available on the market. The author addresses two major issues that represent barriers to effective virus control: 1) there is a high genetic and antigenic variability of BVDV viral subgenotypes and 2) the viral infection of pregnant cows leads to persistently infected cattle which then becomes a reservoir for viral spread. The study provides evidence for potential protection of cattle infected with BVDV expressing the specific combination of E2 proteins utilized in the study and implies that this method of vaccine production could be useful prototype for targeted virus control. This study will be useful to cattle/sheep farmers as well as the meat industry. There are a few minor issues that need to be addressed.

1.     Introduction

The authors did not introduce the E2 protein or describe its role and location in the virus capsid. The vaccines that have been developed so far, predominantly targets the E2 glycoprotein, there is some description of its variability in the different subgenotype but no information on the % sequence variability and the location of the vaccine target. There is also reference to E2 dimerization domain or ectodomain which was not previously described in this document.

2.      Results

In Section 2.2 - Fig. 1 it is not clear what the authors are describing in the FACs analysis. For CHO-1A to 1D do the peaks represent the cells only and the gate shown as P3 represent the cells that are GFP positive or is the final figure labeled 001-GFP a representative of all the positive clones that are expressing GFP?

In Section 2.3, Fig. 3 The authors performed the dosage experiments in sheep but continued with the remaining experiments in cattle. Is the humoral response in the sheep representative of what is expected in the cattle?

Comments on the Quality of English Language

Author Response

Comment 1. Introduction

The authors did not introduce the E2 protein or describe its role and location in the virus capsid. The vaccines that have been developed so far, predominantly targets the E2 glycoprotein, there is some description of its variability in the different subgenotype but no information on the % sequence variability and the location of the vaccine target. There is also reference to E2 dimerization domain or ectodomain which was not previously described in this document.

Response 1. Corrected in the manuscript, lines 75-83. The dimerization domain refers to one of our studies, where we made a chimeric protein using two distinct E2 sequences fused to the Fc region of human IgG [1]. This study aimed to establish a dimer between the Fc segment when expressed in mammalian cells for better E2 exposure.

Comment 2. Results

In Section 2.2 - Fig. 1 it is not clear what the authors are describing in the FACs analysis. For CHO-1A to 1D do the peaks represent the cells only and the gate shown as P3 represent the cells that are GFP positive or is the final figure labeled 001-GFP a representative of all the positive clones that are expressing GFP?

Response 2. We made a nomenclature mistake. 001-GFP is CHO-1E. Corrected in Figure 1.

Comment 3. In Section 2.3, Fig. 3 The authors performed the dosage experiments in sheep but continued with the remaining experiments in cattle. Is the humoral response in the sheep representative of what is expected in the cattle?

Response 3. The humoral response in sheep is indeed representative of that expected in cattle because a high humoral response correlation between sheep and cows when assayed with several inactivated BVDV vaccines has been observed [2]. Added in the manuscript, lines 160-163.

Reviewer 2 Report

Comments and Suggestions for Authors

This paper describes the development of a subunit vaccine prototype against Bovine Viral Diarrhea Virus. The paper is well written in general, and it can be considered for acceptance after addressing the following questions. 

1) Why equal amount of different E2 viral protein was mixed in the study? Based on the Table 1, it seems different serotype has different prevalence. 

2) What was the E2 protein purity after one-step IMAC purification? What was the content of host cell protein, host cell DNA and other impurities? Were there impurities affecting the immunogenicity?

3) In Figure 3, the humoral immune response was studied as E2 protein mixture. How were the results of individual E2 protein?

Author Response

Comment 1. This paper describes the development of a subunit vaccine prototype against Bovine Viral Diarrhea Virus. The paper is well written in general, and it can be considered for acceptance after addressing the following questions. 

1) Why equal amount of different E2 viral protein was mixed in the study? Based on the Table 1, it seems different serotype has different prevalence.

Response 1. The prevalence of the viral subtype and the induction of an immune response with a subunit vaccine candidate are different issues. One of the factors affecting protein immunogenicity is the dose, referring to the amount of protein in a given preparation. There is a dose of antigen above or below which the immune response may not be optimal. Hence, the amount of protein in a recombinant subunit vaccine candidate is crucial to induce a proper immune response. As our recombinant subunit vaccine candidate contains five recombinant proteins, and we do not know the optimal amount of every protein to cause an optimal immune response, we decided to add an equal amount of each protein to induce a reproducible pattern of the immune response. The result of the figure 4D shows the individual humoral immune response against every antigen. If an uneven amount of protein is formulated into the vaccine candidate, inconsistencies in the immune response could be generated during every immunization.

Another concern is achieving consistency in the vaccine production. Formulating an equal amount of each protein into the vaccine candidate guarantees reproducibility during lot-to-lot production. Regarding the viral prevalence, once the animals are appropriately immunized with a multivalent vaccine, they should be protected against any of the five circulating subgenotypes, whether the viral prevalence is high or low.

Comment 2. What was the E2 protein purity after one-step IMAC purification? What was the content of host cell protein, host cell DNA and other impurities? Were there impurities affecting the immunogenicity?

Response 2. The purity of the five E2 proteins was above 60%. When preparing the vaccine formulation, we considered the E2 concentration among the whole sample. Therefore, the E2 amount reported in the final formulation for in vivo assay only refers to E2 proteins. It is known that the humoral response is induced against the whole sample (including impurities). However, the main humoral response would be directed to the most immunogenic and concentrated proteins. In this case, the E2 proteins. Moreover, the neutralization assay demonstrated specific neutralizing antibodies against the BVDV strain, indicating the induction of a properly humoral response.

Comment 3. In Figure 3, the humoral immune response was studied as E2 protein mixture. How were the results of individual E2 protein?

Response 3. The objective of the experiment in sheep was to select the minimal vaccine dose. Thus, we did not evaluate the response against individual E2 antigens in this assay. We only did that in the final experiment in cattle. Considering the immune response correspondence of sheep and cattle against BVDV inactivated vaccines [2], we decided to do the dosage experiment in sheep to select a proper dose for cattle. Despite sheep and cattle have considerable differences in size and body weight, we used the minimal dose obtained in the sheep´s experiment because of the significant differences compared to the negative control.